# Drug Cost Avoidance Resulting from Participation in Clinical Trials: A 10-Year Retrospective Analysis of Cancer Patients with Solid Tumors

**DOI:** 10.3390/cancers16081529

**Published:** 2024-04-17

**Authors:** Maria-Josep Carreras, Berta Renedo-Miró, Carolina Valdivia, Elena Tomás-Guillén, Anna Farriols, Laura Mañós, Jana Vidal, María Alcalde, Isabel De la Paz, Inés Jiménez-Lozano, Maria-Eugenia Palacio-Lacambra, Nuria Sabaté, Enriqueta Felip, Elena Garralda, Margarita Garau, Maria-Queralt Gorgas, Josep Monterde, Josep Tabernero

**Affiliations:** 1Pharmacy Department, Vall d’Hebron University Hospital, E-08035 Barcelona, Spainlauramanos57@gmail.com (L.M.); jana.vidal@vallhebron.cat (J.V.); malcalder@iconcologia.net (M.A.); isabel.delapaz@alirahealth.com (I.D.l.P.); ijimenezl@csapg.cat (I.J.-L.); mariaeugenia.palacio@vallhebron.cat (M.-E.P.-L.); nuriasabate@iconcologia.net (N.S.); mariaqueralt.gorgas@vallhebron.cat (M.-Q.G.); 2Asserta Global Healthcare Solutions, Sant Quirze del Vallés, E-08192 Barcelona, Spain; 3Medical Oncology Department, Vall d’Hebron University Hospital, Vall d’Hebron Institute of Oncology (VHIO), E-08035 Barcelona, Spain; efelip@vhio.net (E.F.); egarralda@vhio.net (E.G.); jtabernero@vhio.net (J.T.)

**Keywords:** clinical trial, solid cancer tumors, drug cost avoidance, pharmaceutical expenditure, sponsor, academic research

## Abstract

**Simple Summary:**

In the present framework of constraints on healthcare budgets, an assessment of costs of antineoplastic drugs is a crucial step in contributing to the sustainability of public-payer systems. This study evaluated the characteristics of cancer clinical trials and potential drug cost avoidance in a large population of adult cancer patients with solid tumors enrolled in clinical trials over a 10-year period (2010–2019). These trials were conducted at the Medical Oncology Department of Vall d’Hebron University Hospital in Barcelona (Spain), one of the largest tertiary care centers in the country. Based on the data of 2930 clinical trials with 10,488 participants, it was found that the total cost of antineoplastic drugs supplied by sponsors in the clinical trials setting was EUR 107,306,084, with a potential cost savings of EUR 92,662,609. Participation in sponsored clinical trials in which drugs are provided free of charge yields considerable cost savings, with benefits in clinical strategies to reduce drug expenditures.

**Abstract:**

The objective of this single-center retrospective study was to describe the clinical characteristics of adult patients with solid tumors enrolled in cancer clinical trials over a 10-year period (2010–2019) and to assess drug cost avoidance (DCA) associated with sponsors’ contributions. The sponsors’ contribution to pharmaceutical expenditure was calculated according to the actual price (for each year) of pharmaceutical specialties that the Vall d’Hebron University Hospital (HUVH) would have had to bear in the absence of sponsorship. A total of 2930 clinical trials were conducted with 10,488 participants. There were 140 trials in 2010 and 459 in 2019 (228% increase). Clinical trials of high complexity phase I and basket trials accounted for 34.3% of all trials. There has been a large variation in the pattern of clinical research over the study period, whereas, in 2010, targeted therapy accounted for 79.4% of expenditure and cytotoxic drugs for 20.6%; in 2019, immunotherapy accounted for 68.4%, targeted therapy for 24.4%, and cytotoxic drugs for only 7.1%. A total of four hundred twenty-one different antineoplastic agents were used, the variability of which increased from forty-seven agents in 2010, with only seven of them accounting for 92.8% of the overall pharmaceutical expenditure) to three hundred seventeen different antineoplastic agents in 2019, with thirty-three of them accounting for 90.6% of the overall expenditure. The overall expenditure on antineoplastic drugs in clinical care patients not included in clinical trials was EUR 120,396,096. The total cost of antineoplastic drugs supplied by sponsors in a clinical trial setting was EUR 107,306,084, with a potential DCA of EUR 92,662,609. Overall, clinical trials provide not only the best context for the progress of clinical research and healthcare but also create opportunities for reducing cancer care costs.

## 1. Introduction

Timely and accurate integration of scientific advances in evidence-based practice is a continuing challenge for the oncology clinician, given the increasing number of new drug approvals and expanded indications for previously approved drugs. All currently available antineoplastic therapies are developed through the clinical trial process led by the pharmaceutical industry and academic research [1]. Commercially sponsored clinical trials are responsible for the development of new anti-cancer drugs, although commercial interests and market expectations do not always coincide with clinical needs. Academic research optimizes the use of drugs in a clinical setting and develops new drug combinations with a strong focus on patients and their unmet needs. Based on clinical trials registered at `ClinicalTrials.gov’, 65% of trials are being supported by health systems, governmental funds, university bodies, and other research organizations [2]. Data from the European Clinical Trials Database (EudraCT) for the period 2005–2013 show that non-commercial sponsors conducted an important proportion (39%) of clinical trials [2]. However, recent statistics (2005–2022) have shown that the percentage of non-commercial trials has decreased to 20% [3], a dangerous trend that will ultimately prove detrimental to cancer care. Moreover, progress in other areas, such as radiation therapy and surgery, can only be achieved by research conducted almost exclusively in academic centers, so the overall contribution of non-commercial institutions to clinical research would be much higher.

Patients benefit from clinical trials due to access to experimental treatments when no other options exist, and to new therapies not yet available, while contributing to the advancement of medical research. In the framework of increasing concerns about the costs of cancer care, data on the cost savings derived from academic clinical trials is limited. Case–control comparisons have shown that participation in cancer clinical trials did not result in significant increases in the direct costs of medical care [4,5,6,7,8,9], but the provision of investigational drug services at academic institutions accounted for substantial drug cost avoidance (DCA) in different disease categories and therapeutic areas, including oncology [10,11,12].

In an analysis of 88 clinical trials with cancer patients in 11 hospitals in Germany from 2002 to 2005, an actual cost saving of EUR 1.5 million was reported [13]. In a lung cancer unit of the Italian National Institute for Cancer Research, the enrolment of 44 patients in 12 sponsored clinical trials allowed the saving of 30% of the pharmaceutical expenses for anti-neoplastic agents in 2010, with additional income from the grants received for each enrolled patient [14]. In a study of 17 phase III clinical trials and 3195 patients conducted by the NCIC Clinical Trials Group in Canada from 1999 to 2011, the total DCA was estimated at CAD 27,952,512, of which targeted therapy constituted 43% [15]. In a study of 357 oncology patients treated on 53 different trial protocols in 2009 and 2010 in a single UK institution, there was an average treatment cost saving of GBP 885,275, largely attributable to pharmaceutical company provision of free drug supplies [16]. In a review of five phase III multicenter, international clinical trials with currently marketed drugs in 136 prostate cancer patients, a total cost avoidance of EUR 696,002 (EUR 5118 per patient) during the study period (1996–2013) was estimated [17]. In another study of 89 breast cancer patients included in 37 clinical trials between 2014 and 2016, the use of an investigational drug as compared to the best standard of care treatments was associated with a total cost avoidance of EUR 957,246 (EUR 10,756 per patient) [18]. In a single Spanish National Health System institution, the overall cost of treatment for 68 adult cancer patients enrolled in 20 different clinical trials was 79% lower in comparison to the routine standard of care, with an average treatment cost per patient and protocol ranging from an excess of EUR 8193 to a saving of EUR 59,770 [19].

However, real-world evidence of drug cost avoidance (DCA) from a large number of sponsored clinical trials in cancer patients is still lacking. Therefore, the objective of the study was to analyze the pharmaceutical antineoplastic expenditure and potential drug cost savings in a large cohort of adult cancer patients with solid tumors enrolled in sponsored clinical trials in a single institution over a 10-year period. The analysis of data by phases of the trials, pharmacological drug classes, tumor location, and active drugs is a differential contribution of the study. Assessment of drug cost savings in a very large cohort of patients with solid tumors will contribute to removing concerns regarding the additional costs of clinical trial settings.

## 2. Materials and Methods

### 2.1. Design and Patients

This was a single-center retrospective study of adult cancer patients with solid tumors enrolled in any ongoing clinical trial over a 10-year period (2010–2019) in the Medical Oncology Department of Vall d’Hebron University Hospital (HUVH) in Barcelona, Spain. Data on the pharmaceutical expenditure of adult cancer patients with solid tumors treated at HUVH during the same 10-year period in the health care setting, that is, outside the clinical trial setting, were previously reported [20].

The hospital is the largest tertiary care center in Catalonia and one of the largest in the country. It belongs to the network of medical centers of the national public health system, which provides healthcare services free of charge to all Spanish citizens independently of their demographic characteristics, socioeconomic status, or tax contribution level. In the case of oncological patients, antineoplastic treatments are reimbursed to the hospital by the public health authority. The HUVH has a reference population of 430,000 inhabitants and an annual budget of EUR 630 million [21]. The Medical Oncology Department develops extensive clinical and research activities in coordination with the Vall d’Hebron Institute of Oncology (VHIO). The Pharmacy Department provides services to the general, traumatology, and pediatric/maternal areas, and integrates the Outpatient Prescriptions Unit and the Pharmacy Oncology Unit with various subunits located at the different oncology day hospitals.

Inclusion criteria were: patients of both genders, aged 18 years or older, diagnosed with malignant solid tumors, enrolled in a clinical trial, and treated with any specific antineoplastic drug by the parenteral or oral routes. The parenteral route included intravenous, intramuscular, and subcutaneous drug delivery. Patients with solid tumors not enrolled in a clinical trial were excluded. Approval by the Clinical Research Ethics Committee was waived because of the quality of care nature of the study.

### 2.2. Study Variables

For each patient, the following variables were recorded: phase of the clinical trial including phase 0; phase I and basket trials; phase II (tumor-specific); phase III; post-authorization and rollover studies; cancer type, classified by body location/system following the National Cancer Institute classification [22]; and pharmacological classes of antineoplastic drugs categorized following the International Common Denomination of Drugs (ICD) or International Non-Proprietary Names (INN) or the Spanish Agency of Medicines and Medical Devices [23], grouped into cytotoxic drugs, immunotherapy, targeted therapy, and other active drugs. For these subgroups, the number of patients and pharmaceutical expenditures for each natural year of the study period were recorded depending on the availability of data.

### 2.3. Pharmaceutical Expenditure, Sponsor Contribution, and Drug Cost Avoidance

The data for the sample of patients enrolled in clinical trials during the 2010–2019 study period were obtained from the Fundanet (Cantabria, Spain) CTMS (Clinical Trials Management) software (V.2020-13) (https://vhio.fundanetsuite.com/EstudiosClinicos/ (accessed 20 June 2020)) of the VHIO Center of Clinical Trials. Data extracted from this database were imported to the Microsoft Excel program to obtain the following information for each study year: number of clinical trials actively recruiting patients; number of clinical trials with actively recruited patients categorized by phases and cancer type; number of recruited patients by trial phases; and number of patients on active treatment by trial phases.

The costs of all pharmaceutical specialties provided by the sponsors were considered whenever they were marketed drugs at the time of prescription. In randomized double-blind clinical trials, in which the Pharmacy Service was masked regarding the identification of the investigational drug before opening the randomization code, the costs of the doses were calculated by applying the allocation ratio to the number of randomized arms of the protocol. Two types of economic analysis were performed:

(1) The economic contribution of the sponsor in the provision of drugs to carry out clinical trials was calculated according to the actual cost (for each year) of the pharmaceutical specialties that the Vall d’Hebron University Hospital would have had to purchase to carry out these clinical trials in the absence of sponsorship. Details of the calculation of pharmaceutical expenditure and the direct cost of acquiring the pharmaceutical specialties of antineoplastic drugs were previously reported [20].

(2) To obtain the estimated potential annual DCA in the total population of the clinical trials, the mean cost of antineoplastic treatment per patient and year, in each tumor location, of the population of patients treated in our center in the healthcare setting, that is, not included in clinical trials, in the same period (2010–2019), was previously calculated. For this calculation, data were used from the exhaustive analysis of pharmaceutical expenditure involved in antineoplastic treatment of the healthcare population previously reported in our previous study [20]. This mean cost obtained per patient in the healthcare setting outside of clinical trials (in each year and each tumor location) was applied to the number of participants in clinical trials to obtain an estimate of the pharmaceutical expenditure that these patients would have incurred with the treatment that would have been prescribed and used in these patients on the same dates if they had been treated in our healthcare population. The potential annual DCA by tumor location was calculated for clinical trial participants treated with intravenous antineoplastic medications only, as data on tumor locations in patients receiving oral antineoplastic medications in the clinical trial were not available in the anonymized database.

### 2.4. Statistical Analysis

The data are reported as descriptive statistics. Categorical variables are expressed as frequencies and percentages, and quantitative variables as the mean and standard deviation (SD) or the median and interquartile range (IQR) (25th–75th percentile) if non-normal distributions of data. Analyses were performed using the Pharmacy Analytics Manager (V.2022.1.0) (https://www.asserta.net (accessed on 20 July 2020)).

## 3. Results

### 3.1. Clinical Trials and Patient Population

A total of 2930 clinical trials with active recruitment were conducted at the Medical Oncology Department between 2010 and 2019 for solid tumor treatment. There were 140 in 2010 and 459 in 2019, an increase of 228%. As shown in Table 1, clinical trials of high complexity phase I and basket trials accounted for 34.3% of all the trials. Also, clinical trials focused on a specific tumor type accounted for the largest percentage (76.5%), but trials not involving a specific tumor type increased markedly from 36 in 2010 to 103 in 2019, an increase of 186%.

Between 2010 and 2019, specific antineoplastic treatment was delivered in the clinical trial setting to 10,488 patients, 4964 (47.3%) of whom were treated with intravenously administered drugs, and 5524 (52.7%) with oral drugs. The number of patients on active treatment in the different phases of the clinical trials between 2016 and 2019 is shown in Figure 1.

### 3.2. Pharmaceutical Expenditure on Cancer Treatment in Clinical Trials (Financial Contribution from the Sponsor)

The overall pharmaceutical expenditure during the study period (2010–2019) based on the contribution of the sponsor was EUR 107,306,084, divided into EUR 72,266,379 for treatment with intravenous antineoplastic drugs, and EUR 35,039,705 for oral drugs (Table 2).

### 3.3. Expenditure by Pharmacological Classes

Expenditures by pharmacological classes and types of antineoplastic drugs were available for the period between 2013 and 2019. The overall use of intravenous and oral antineoplastic drugs accounted for 94% of the expenditures (EUR 100,944,351). As shown in Table 3, targeted therapy was associated with the largest expenditure (58%), followed by immunotherapy (35.3%), and cytotoxic drugs (6.7%). There was a marked increase in expenditures from a total of EUR 5,177,221 in 2013 to EUR 27,572,327 in 2019. However, annual increases varied from 5.1% (2014 to 2015) to 92.1% (2016–2017) and 6% (2018–2019).

The data on pharmaceutical expenditure by pharmacological classes for the complete 10-year study period were available for intravenous antineoplastic agents (Table 4). There was a large variation in the pattern of clinical research over the study period: in 2010, targeted therapy accounted for 79.4% of expenditure and cytotoxic drugs for 20.6%; in 2019, immunotherapy accounted for 68.4%, targeted therapy for 24.4%, and cytotoxic drugs for only 7.1%. Also, the percentages of patients treated with these pharmacological classes in 2010 and 2019 showed important changes, from 75.2% to 33.8% for cytotoxics, 0.8% to 67.2% for immunotherapy, and 54.2% to 28% for targeted therapy.

### 3.4. Expenditure by Antineoplastic Agent

The antineoplastic agents used for treating patients included in clinical trials between 2010 and 2019 are shown in Appendix A. A total of four hundred twenty-one different antineoplastic agents were used, the variability of which increased from forty-seven agents in 2010, with only seven of them accounting for 92.8% of the overall pharmaceutical expenditure) to three hundred seventeen different antineoplastic agents in 2019, with thirty-three of them accounting for 90.6% of the overall expenditure (Table 5).

### 3.5. Expenditure by Tumor Location

In 2010, patients with 17 tumor locations received intravenous antineoplastic drugs, with five tumor sites (breast, colon, ovary, non-small cell lung cancer, and prostate) accounting for 90% of all cancer cost treatment, but this number increased to 30 tumor locations, with fifteen tumor sites, in particular non-small cell lung cancer, advanced solid tumors, breast, colon, and ovary accounting for 90% of expenditure in 2019 (Table 6). Also, the highest percentages of patients treated in 2019 corresponded to advanced solid tumors (20.2%) and non-small cell lung cancer (17.5%).

The pharmaceutical expenditures by tumor location for each study year in patients enrolled in clinical trials treated with intravenous antineoplastic agents are shown in Appendix A.

### 3.6. Estimation of Potential Drug Cost Avoidance

The estimated potential drug cost of antineoplastic treatments avoided for patients included in clinical trials during the study period is shown in Table 7. The total pharmaceutical expenditure avoided was EUR 92,662,609, divided into EUR 34,463,891 for the use of antineoplastic agents administered intravenously, and EUR 58,198,718 for those given by the oral route.

The mean cost obtained per patient in the healthcare setting outside of clinical trials (in each year and each tumor location) is presented in Appendix A. Applying this mean cost per patient to each patient included in a clinical trial allowed us to obtain a distribution of potential DCA by tumor location and study years, shown in Appendix A (for intravenous drugs only). Breast cancer, rectal cancer, non-small cell lung cancer, and melanoma were the tumor locations in the upper rank of DCA.

To put into context what the savings in clinical trials have represented for our institution, previously reported data on pharmaceutical expenditure on antineoplastic drugs for patients with solid tumors treated in the healthcare setting (outside clinical trials) at our center are presented (EUR 120,396,096) for the same study period 2010–2019 [20]. The total cost of antineoplastic drugs supplied by sponsors in the clinical trials setting was EUR 107,306,084, with a potential DCA of EUR 92,662,609 (Table 8).

Figure 2 shows the percentages of pharmaceutical expenditure on antineoplastic drugs for the treatment of patients with solid cancers during the study period in the two populations of patients treated in the routine healthcare setting (mean cost 66%, range 41% to 82%) and participants in clinical trials with sponsor financing (mean cost 44%, range 18% to 59%). On the other hand, the economic contribution of the sponsors accounted for 89% of the expenditure incurred by HUVH for treating cancer patients with antineoplastic drugs in the routine healthcare setting.

Figure 3 shows the percentages of pharmaceutical expenditure on antineoplastic drugs for solid cancers during the study period in the routine healthcare setting (mean cost 62%, range 49% to 60%) and the potentially avoided expenditure due to the performance of clinical trials (mean cost 48%, range 40% to 51%). Also, DCA accounted for 77% of the expenditure incurred by HUVH for treating cancer patients with antineoplastic drugs in the routine healthcare setting.

## 4. Discussion

The Vall d’Hebron Institute of Oncology (VHIO) is an oncology center of excellence and innovation, in which investigators and clinicians follow a translational research model working in multidisciplinary teams focused on advancing in the field of personalized and specific therapies against cancer. Scientific activity at the VHIO is based on three main programs, including preclinical research, translational research, and clinical research and transversal technologies. Each program incorporates high-level research teams dedicated to ensuring that research results can be implemented and translated into tangible benefits for patients (https://vhio.net). During the study period of 2010–2019, we identified a high number of clinical studies for the treatment of solid tumors with patients being recruited or actively participating in clinical trials. In 2019, there were 459 clinical trials with ongoing recruitment, 162 (35.3%) of which were phase I and basket trials. Also, in 2019, the total number of 1962 patients treated in clinical trials accounted for 41.4% of all cancer patients with solid tumors treated at the Medical Oncology Department, 38.4% participating in 162 phase I and basket trials. According to the Spanish Registry of Clinical Studies (REEC), in August 2020, 597 clinical cancer trials in adults with active recruitment were registered in this database, 188 (31.5%) of which were phase I trials [24]. These data show the leadership role of VHIO and HUVH in the overall and complex cancer clinical research landscape in Spain.

In the 2010–2019 study period, a total number of 10,488 patients were included in clinical trials, with treatments that resulted in drug contributions by sponsors for our center of EUR 107,306,084. The potential estimated DCA, according to our approximation, is EUR 92,662,609. In all the cases, the drug cost estimations were based on the same actual purchase price for the HUVH of the pharmaceutical specialties corresponding to the doses prescribed at the time of each prescription. The lower amount obtained from the calculation of the DCA is explained by the lower drug cost of antineoplastics used in routine healthcare conditions (used for the calculation of the DCA) as compared to innovative high-cost therapies in clinical trials (economic contribution of the sponsor). The distribution of expenses in each of the two populations clearly indicates greater use of innovative and high-cost therapies in the clinical trial population compared to the routine healthcare population; the expenses in targeted therapy plus immunotherapy were 93.31% compared to 81.37%, and in cytotoxic plus other drugs were 6.69% compared to 18.63%, respectively [20]. Also, the economic contribution of sponsors and DCA corresponded to 89% and 77%, respectively, of the total pharmaceutical expenditure of HUVH in antineoplastic agents for treating cancer patients with solid tumors in the routine healthcare setting over the 10-year study period.

Academic research has been considered a source of sustainability of public healthcare systems [25,26], with trial sponsors providing tangible benefits [27,28]. However, quantification of DCA in cancer patients treated in clinical trials has not been consistently documented, particularly due to deficient infrastructure information systems of hospitals, and because trial drugs provided by sponsors are not systematically included in pharmacoeconomic analyses of cancer medications [29]. Previous studies addressing the cancer treatment costs of participants in cancer clinical trials have been carried out in the framework of healthcare systems in the USA, Canada, Germany, the UK, Italy, Taiwan, and Spain [10,11,12,13,14,15,16,17,18,30]. All of these studies have documented economic benefits for the medical center when patients participating in clinical trials receive drugs financed by external sources, although the results are difficult to compare due to differences in the design of the studies and methodological heterogeneity in the analysis of avoided costs, such as estimations of expenditures of commercialized or non-commercialized investigational drugs only, the regimen of the control arm, the standard of care treatment, and cost allocation of drugs with prices from different sources, etc.). Our study has made two estimates: that of the cost to the HUVH of the drugs provided by the promoter (which would coincide with the strategy of Calvin-Lamas et al. [17]) and would be interpreted as an avoided cost for the hospital if the center financed the studies carried out, and the cost avoided in the treatment of patients included in a clinical trial (DCA) if these patients were treated like routine healthcare patients. This strategy represents a more realistic approach to care practice than the consideration of theoretical costs of the standard alternative or the control arms of the trials [10,11,13,14,17,18]. Our study has other distinctive features compared with previous publications, including the 10-year study period, the large number of patients, the inclusion of different phases of the trials, and the analysis by pharmacological classes, route of administration, tumor location, and individual antineoplastic agents.

The pattern of clinical research by pharmacological classes showed an important variation over the 10-year study period, with the proportion of cytotoxic drugs decreasing from 20.6% in 2010 to 7.1% in 2019 and targeted therapy from 79.4% to 24.4%, whereas immunotherapy accounted for 68.4% in 2019. In the study by Shen et al. [12] conducted in 2008 at the most prominent medical center in Taiwan, targeted therapy accounted for 84.1% of the total cost avoidance of anti-cancer trials, with lapatinib as the most prominent drug. In the analysis of phase III trials between 1999 and 2011 by Tang et al. [15], targeted therapy constituted 43% of total DCA.

In patients treated with intravenous antineoplastic drugs, a total of 40 tumor locations were treated with 421 different antineoplastic drugs, which indicates the high representatively of the clinical trial setting in our clinical practice. A total of 15 cancer sites accounted for 90% of the economic contribution of sponsors, particularly non-small cell lung cancer, advanced solid tumors, and breast, colon, and ovarian cancer.

In fact, most of the previous studies on clinical trials analyzed a single tumor site, such as lung cancer and mesothelioma [14], prostate [17], breast [18], or cancer in general, without specification of the organ affected. Bredin et al. [30] evaluated 37 clinical trials grouped into five tumor groups (lung, gynecology, neurology, hematology, and genitourinary) and showed a large range of cost avoidance per patient due to a combination of factors, including the duration patients remain on the study (which varies widely among the tumor groups), the number of days in each cycle a drug is administrated, and the price of the drugs. Of the 17 clinical trials in the study by Tang et al. [15], most trials involved breast, lung, and ovarian tumors. In our study, in 2019, clinical trials with intravenous and oral antineoplastic agents were analyzed, with the inclusion of a total of 1122 different patients, which accounted for 25.8% of gastrointestinal tumors, 23.3% of lung cancer, 13% of breast tumors, 11.8% of gynecologic tumors, and 9.5% of genitourinary tumors. The antineoplastic agents representing 50% of the financial contribution of sponsors were pembrolizumab (13%), nivolumab (9%), cabozantinib (8.7%), atezolizumab (7.2%), bevacizumab (6.2%), and palbociclib (5.3%).

The fact that we have not considered the contribution, or costs avoided, of the supportive therapy is a limitation in the study. According to Bredin et al. [30], antiemetics and colony-stimulating growth factors account for 2.9% of costs regarding conventional cytotoxics, and anti-diarrheal and antihistamines for 0.28% regarding targeted therapy. In our previous study [20], supportive therapy (antiemetics, erythropoietins, colony-stimulating growth factors, and hypocalcemia medications) accounted for 4.6% of the total pharmaceutical expenditure for the treatment of solid tumors in patients treated in the routine healthcare setting, and although this percentage is low, it showed a decreasing trend along the years, with only 1.4% in 2019. Other aspects involved in the costs of the clinical trials, such as administrative or infrastructure variables, were not evaluated.

Regarding differences between commercial and non-commercial clinical trials, extrapolation of results from healthcare systems with different public or private financial structures is a complex issue. In an analysis of the financial sustainability of an oncology clinical trial unit from a Canadian medical center between 2007 and 2011, the authors describe the negative income of the trials promoted by cooperative groups due to not-funded or underfunded follow-up costs, so the unit accrued more patients in new clinical trials or existing industry-funded trials to offset that deficit, but as the number of patients on follow-up increased, the fiscal deficit grows larger each year, perpetuating the cycle [31]. Although patients included in clinical trials receive more frequent clinical visits, which translate into a higher number of tests and procedures, those receiving the routine healthcare standard of care (SOC) also receive costly therapies [8], and the differences disappear due to the free provision of antineoplastic drugs in clinical trials by the sponsors, which accounts for the largest difference in pharmaceutical expenditure between the two populations [9]. In a retrospective cost analysis of 68 adult cancer patients treated in 20 different clinical trials in a single Spanish institution, it was detected that although the load of procedures was 32% higher, the costs for the pharmacological treatment system were 79% lower in the cohort of clinical trial patients, and it was concluded that in the entirety of the clinical trial units, the most favorable scenario is a balanced portfolio of both commercial and non-commercial trials [19].

As a relevant limitation of our study, we do not assess the general costs of the administrative infrastructure for the execution of clinical trials, nor of the costs of all medical procedures (e.g., medical visits, management of adverse effects, lack of efficacy of treatments carried out, etc.). Also, other factors contributing to further analysis of the meaningfulness of cost savings from enrollment in clinical trials having access to free medications, such as patients’ demographic characteristics or their socio-economic status, were not evaluated. These objectives were not in the scope of our work, which does not aim to evaluate the sustainability of the clinical trials unit (financed mostly by the private resources generated by the trials themselves) but aims to contribute knowledge to the expenditure contributed in drugs by the promoters and to the expenditure potentially avoided in the specific subject of pharmacological treatment from the perspective of reimbursement by the national health system. The study of the economic burden represented by clinical trials has been carried out from the payer’s perspective, that is, the national public health administration because, in our country, all citizens regardless of their social, demographic, or economic status have universal free access to all funded oncological treatments through the Spanish national healthcare system. Overall, clinical trials provide not only the best context for the progress of clinical research and healthcare but also create opportunities for reducing public cancer-care costs [32].

In summary, the added value of the present analysis is based on its relevant differences compared with previous studies [4,5,6,7,8,9,10,11,12,13,14,15,16,17,18,19,30], that is, the assessment of all consecutive cancer patients enrolled in sponsored clinical trials, considering all tumor locations, and treated at the Medical Oncology Department of the largest hospital in Spain in which the largest number of oncological clinical trials are performed in the country. The analysis of 2930 clinical trials with the enrollment of 10,488 patients with solid tumors located at 30 sites showed an impressive DCA of EUR 92,662,609. The detail of the data presented may be of interest for comparative purposes in studies by other authors.

## 5. Conclusions

The pattern of research through oncological clinical trials at the HUVH has allowed early access to innovation and a contribution to knowledge that reverts to clinical care practice. At the same time, it implies a lightening of the economic burden on the healthcare system by the impact of the free supply of investigational drugs and the potential avoided pharmaceutical expenditure from the perspective of reimbursement by the national health system, according to an approximation of the cost of treating patients in the trial setting with the actual clinical practice of the routine healthcare population.

## Figures and Tables

**Figure 1 cancers-16-01529-f001:**
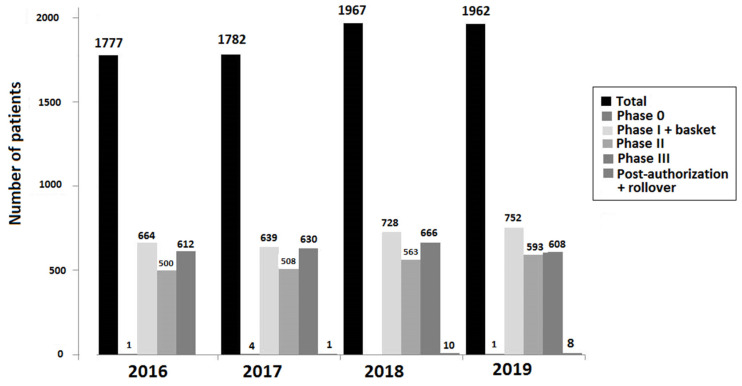
Distribution of 7488 patients on active treatment by phases of trials (2016–2019).

**Figure 2 cancers-16-01529-f002:**
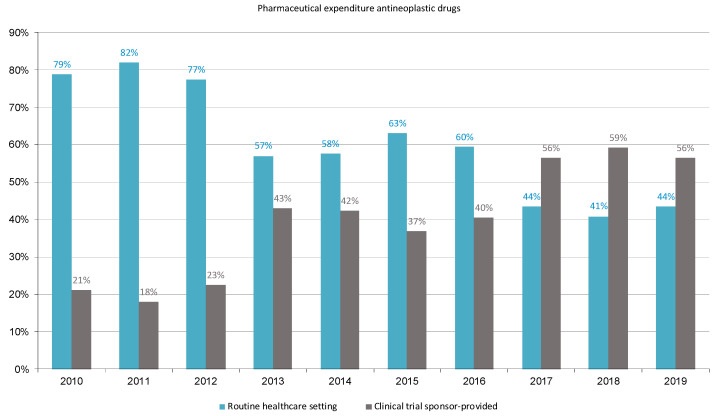
The proportion of pharmaceutical expenditure for antineoplastic drugs in the routine healthcare setting and in sponsor-supported clinical trials.

**Figure 3 cancers-16-01529-f003:**
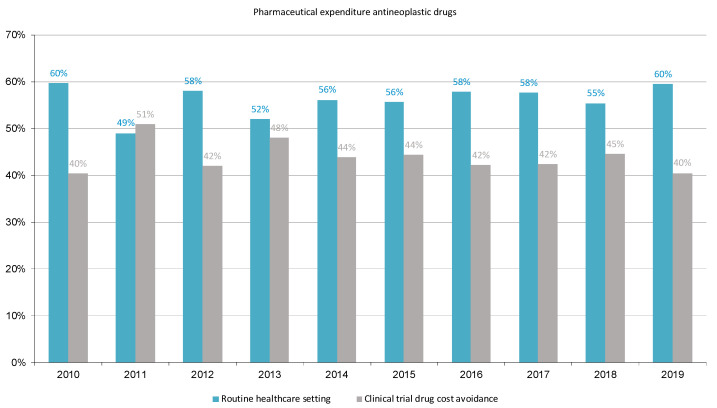
The proportion of pharmaceutical expenditure for antineoplastic drugs in the routine healthcare setting and potentially avoided due to sponsored financial contribution.

**Table 1 cancers-16-01529-t001:** Number of clinical trials conducted at the Medical Oncology Department with active recruitment of patients at any time during the study period.

Study Year	Clinical Trials					
	Total	Phase 0 *n* (%)	Phase I and Basket *n* (%)	Phase II *n* (%)	Phase III *n* (%)	Post-Authorization and Rollovers *n* (%)
2010	140	0	37 (26.4)	54 (38,6)	49 (35.0)	0
2011	161	0	48 (29.8)	57 (35.4)	56 (34.8)	0
2012	219	0	66 (30.1)	85 (38.8)	68 (31.0)	0
2013	232	0	75 (32.3)	96 (41.4)	61 (26.3)	0
2014	251	0	83 (33.1)	99 (39,4)	64 (25.5)	5 (2.0)
2015	303	0	106 (35.0)	94 (31.0)	89 (29.4)	14 (4.6)
2016	370	0	129 (34.9)	117 (31.6)	108 (29.2)	16 (4.3)
2017	374	0	137 (36.6)	107 (28.6)	111 (29.7)	19 (5.1)
2018	421	0	161 (38.2)	131 (31.1)	107 (25.4)	22 (5.2)
2019	459	1 (0.2)	162 (35.3)	141 (30.7)	121 (26.4)	34 (7.4)
2010–2019	2930	1 (0.03)	1004 (34.3)	981 (33.5)	834 (28.5)	110 (3.7)

**Table 2 cancers-16-01529-t002:** Pharmaceutical expenditure by type of antineoplastic drug in the period 2010–2019 in the clinical trial setting.

Study Years	Expenditures of Antineoplastic Drugs	
	Intravenous	Oral
2010	EUR 2,457,850	NR
2011	EUR 1,700,250	NR
2012	EUR 2,203,633	NR
2013	EUR 3,331,912	EUR 1,845,309
2014	EUR 4,636,490	EUR 2,022,064
2015	EUR 4,120,596	EUR 2,876,494
2016	EUR 6,220,320	EUR 3,545,183
2017	EUR 12,675,915	EUR 6,086,139
2018	EUR 17,844,749	EUR 8,166,853
2019	EUR 17,074,664	EUR 10,497,663
Total	EUR 72,266,379	EUR 35,039,705
Overall	EUR 107,306,084	

NR: not registered.

**Table 3 cancers-16-01529-t003:** Pharmaceutical expenditure by pharmacological classes and study year (2013–2019) in the clinical trial setting.

	Cytotoxic Drugs	Immunotherapy	Targeted Therapy	Other Drugs	Total
Antineoplastic drugs					
Intravenous	EUR 6,322,282	EUR 35,623,961	EUR 23,958,403		EUR 65,904,646
Oral	EUR 424,306		EUR 34,609,333	EUR 6066	EUR 35,039,705
Total	EUR 6,746,588	EUR 35,623,961	EUR 58,567,736		EUR 100,944,351
Study year					
2013	EUR 957,915	EUR 590,250	EUR 3,629,056		EUR 5,177,221
2014	EUR 613,360	EUR 1,517,637	EUR 4,527,558		EUR 6,658,554
2015	EUR 796,317	EUR 222,342	EUR 5,978,431		EUR 6,997,090
2016	EUR 891,738	EUR 2,536,316	EUR 6,337,449		EUR 9,765,502
2017	EUR 1,175,345	EUR 6,900,964	EUR 10,685,745		EUR 18,762,054
2018	EUR 1,001,328	EUR 12,169,256	EUR 12,841,019		EUR 26,011,603
2019	EUR 1,310,587	EUR 11,687,195	EUR 14,568,479	EUR 6066	EUR 27,572,327

**Table 4 cancers-16-01529-t004:** Pharmaceutical expenditure by type of intravenous antineoplastic drug in the period 2010–2019 in the clinical trial setting.

Study Year	Cytotoxic Drugs	Immunotherapy	Targeted Therapy	Total Expenditure	Change vs. Previous Year, %
2010	EUR 506,138	-EUR	EUR 1,951,712	EUR 2,457,850	
2011	EUR 599,725	-EUR	EUR 1,100,526	EUR 1,700,250	−30.8%
2012	EUR 860,070	-EUR	EUR 1,343,563	EUR 2,203,633	29.6
2013	EUR 863,237	EUR 590,250	EUR 1,848,425	EUR 3,331,912	51.2
2014	EUR 568,777	EUR 1,517,637	EUR 2,550,076	EUR 4,636,490	39.1
2015	EUR 787,984	EUR 222,342	EUR 3,110,270	EUR 4,120,596	−11.1
2016	EUR 866,817	EUR 2,536,316	EUR 2,817,187	EUR 6,220,320	51.0
2017	EUR 1,047,329	EUR 6,900,964	EUR 4,727,621	EUR 12,675,915	103.8
2018	EUR 940,547	EUR 12,169,256	EUR 4,734,946	EUR 17,844,749	40.8
2019	EUR 1,217,591	EUR 11,687,195	EUR 4,169,878	EUR 17,074,664	−4.3

**Table 5 cancers-16-01529-t005:** Antineoplastic agents used in patients treated in the clinical trial setting in 2010 and 2019.

2010			2019		
Antineoplastic Agent	Expenditure	% of Total Expenditure	Antineoplastic Agent	Expenditure	% of Total Expenditure
Trastuzumab	EUR 919,484	37.4	Pembrolizumab	EUR 3,590,128.10	13.0
Bevacizumab	EUR 449,063	18.3	Nivolumab	EUR 2,489,205.42	9.0
Aflibercept/placebo	EUR 301,600	12.3	Cabozantinib	EUR 2,392,934.10	8.7
Docetaxel	EUR 191,287	7.8	Atezolizumab	EUR 1,983,242.04	7.2
Pemetrexed	EUR 145,919	5.9	Bevacizumab	EUR 1,696,849.55	6.1
Cetuximab	EUR 145,079	5.9	Palbociclib	EUR 1,449,416.28	5.3
Panitumumab	EUR 129,287	5.3	Olaparib	EUR 1,338,767.04	4.9
Total		92.8	Lorlatinib	EUR 1,094,262.59	4.0
			Ipilumumab	EUR 786,563.46	2.8
			Pembrolizumab	EUR 670,891,67	2.4
			Pemetrexed	EUR 615,425.91	2.2
			Pertuzumab	EUR 581,406.00	2.1
			Avelumab	EUR 568,416.00	2.1
			Abiraterone	EUR 500,524.62	1.8
			Pembrolizumab/placebo	EUR 405,841.74	1.5
			Cetuximab	EUR 393,887.10	1.4
			Lenvatinib	EUR 386,416.81	1.4
			Dinutuximab	EUR 348,727.07	1.3
			Atezolizumab/placebo	EUR 326,070.63	1.2
			Osimertinib	EUR 300,501.60	1.1
			Sunitinib	EUR 299,647.57	1.1
			BMS-986213/Nivolumab	EUR 289,906.56	1.0
			Axitinib	EUR 280,783.80	1.0
			Enzalutamide	EUR 274,394.96	1.0
			Trastuzumab s.c.	EUR 271,830.00	1.0
			Apalutamide	EUR 259,110.00	0.9
			Ipilimumab/placebo	EUR 222,314.53	0.8
			Trastuzumab	EUR 217,028.66	0.8
			Alectinib	EUR 204,001.91	0.7
			Panitumumab	EUR 183,674.66	0.7
			Nivolumab/placebo	EUR 183,520.80	0.7
			Niraparib	EUR 181,838.25	0.7
			Olaratumab/placebo	EUR 181,812.21	0.7
			Total		90.6

**Table 6 cancers-16-01529-t006:** Percentages of pharmaceutical expenditures and patients treated in the clinical trial setting with intravenous antineoplastic agents by tumor location in 2010 and 2019.

2010			2019		
Tumor Location	% of Total Expenditure	% of Total Patients	Tumor Location	% of Total Expenditure	% of Total Patients
Breast	46.4	34.8	Lung, non-small cell	27.0	17.5
Colon	18.0	19.1	Advanced solid tumor	10.1	20.2
Ovary	14.3	8.7	Breast	9.5	9.9
Lung, non-small cell	9.6	13.5	Colon	8.0	7.6
Prostate	5.8	3.0	Ovary	7.0	5.2
Pancreas endocrine	1.9	0.8	Melanoma	6.2	5.4
Rectum	1.5	3.2	Kidney	5.2	3.7
Advanced solid tumors	1.3	8.4	Pancreas exocrine	2.9	4.3
Oral cavity and oropharynx	0.4	1.6	Uterine cervix	2.7	3.4
Stomach	0.3	1.6	Lung, small cell	2.6	2.7
Pancreas exocrine	0.2	2.4	Endometrium	2.3	3.2
Lung, small cell	0.2	0.5	Urinary bladder	2.2	2.7
Urinary bladder	0.04	2.2	Stomach	2.2	3.1
Kidney	0.02	0.5	Oral cavity and oropharynx	2.1	2.2
Soft tissue sarcoma	0.01	0.3	Mesothelioma	1.7	1.5
Endometrium	0.01	0.8	Esophagus	1.5	1.1
Extrahepatic bile duct	0.0	0.3	Soft tissue sarcoma	1.4	1.0
			Rhinopharynx/cavum	1.2	0.4
			Neuroendocrine	0.7	1.4
			Hepatic carcinoma	0.7	0.5
			Thymoma	0.7	0.5
			Brain	0.5	0.7
			Prostate	0.5	1.0
			Squamous cell skin cancer	0.4	0.3
			Larynx/hypopharynx	0.3	0.09
			Rectum	0.2	0.6
			Basal cell carcinoma	0.2	0.2
			Merkel carcinoma	0.07	0.2
			Extrahepatic bile duct	0.02	1.1
			Urothelial cancer	0.0	0.09
			Adrenal		0.09
			Anus		0.4
			Pancreas endocrine		0.5
			Thyroid		0.3
			Gallbladder		0.09

**Table 7 cancers-16-01529-t007:** Estimated antineoplastic drug cost avoidance associated with patients included in clinical trials conducted between 2010 and 2019.

Study Years	Expenditures on Antineoplastic Drugs		
	Intravenous	Oral	Total
2010	EUR 1,650,750	EUR 4,541,111	EUR 6,191,865
2011	EUR 1,205,680	EUR 6,829,844	EUR 8,035,524
2012	EUR 1,322,395	EUR 4,141,530	EUR 5,463,925
2013	EUR 2,349,744	EUR 3,994,676	EUR 6,344,420
2014	EUR 2,157,533	EUR 4,933,760	EUR 7,091,293
2015	EUR 3,034,919	EUR 6,474,355	EUR 9,509,274
2016	EUR 4,087,619	EUR 6,370,112	EUR 10,457,731
2017	EUR 4,722,001	EUR 5,911,956	EUR 10,633,957
2018	EUR 7,266,822	EUR 7,196,238	EUR 14,463,060
2019	EUR 6,666,428	EUR 7,805,132	EUR 14,471,560
Total	EUR 34,463,891	EUR 58,198,718	EUR 92,662,609

**Table 8 cancers-16-01529-t008:** Pharmaceutical expenditure for the treatment of patients with solid tumors during the study period (2010–2019).

Study Years	Pharmaceutical Expenditure of Antineoplastic Agents					
	Healthcare Setting		Clinical Trial Setting		Potential Drug Cost Avoidance	
	Intravenous	Oral	Intravenous	Oral	Intravenous	Oral
2010	EUR 6,565,263	EUR 2,579,163	EUR 2,457,850	NR	EUR 1,650,750	EUR 4,541,115
2011	EUR 4,464,740	EUR 3,271,684	EUR 1,700,250	NR	EUR 1,205,680	EUR 6,829,844
2012	EUR 4,695,722	EUR 2,853,205	EUR 2,203,633	NR	EUR 1,322,395	EUR 4,141,530
2013	EUR 4,436,132	EUR 2,437,685	EUR 3,331,912	EUR 1,845,309	EUR 2,349,744	EUR 3,994,676
2014	EUR 5,770,959	EUR 3,302,873	EUR 4,636,490	EUR 2,022,064	EUR 2,157,533	EUR 4,933,760
2015	EUR 7,596,964	EUR 4,346,928	EUR 4,120,596	EUR 2,876,494	EUR 3,034,919	EUR 6,474.355
2016	EUR 9,769,375	EUR 4,581,945	EUR 6,220,320	EUR 3,545,183	EUR 4,087,619	EUR 6,370,112
2017	EUR 9,475,022	EUR 4,990,955	EUR 12,675,915	EUR 6,086,139	EUR 4,722,001	EUR 5,911,956
2018	EUR 11,252,776	EUR 6,723,301	EUR 17,844,749	EUR 8,166,853	EUR 7,266,822	EUR 7,196,238
2019	EUR 12,666,317	EUR 8,615,087	EUR 17,074,664	EUR 10,497,663	EUR 6,666,428	EUR 7,805,132
Total	EUR 76,693,270	EUR 43,702,826	EUR 72,266,379	EUR 35,039,705	EUR 34,463,891	EUR 58,198,718
Overall	EUR 120,396,096		EUR 107,306,084		EUR 92,662,609	

NR: not registered.

## Data Availability

Study data are available from the corresponding author upon reasonable request.

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
