# Peer review of "Drug Cost Avoidance Resulting from Participation in Clinical Trials: A 10-Year Retrospective Analysis of Cancer Patients with Solid Tumors"

_cancers, 2024, doi:10.3390/cancers16081529_

Round 1

Reviewer 1 Report

Comments and Suggestions for Authors

The authors measured the potential anti-cancer medication cost savings for 10,488 adult cancer patients with solid tumors, enrolled in 2930 clinical trials between 2010 and 2019, at the Medical Oncology Department of one of the largest tertiary University Hospital in Barcelona of Spain.

I understand that the authors intended to assess how the clinical trials can help to relief the financial concerns of cancer patients for medication with those expensive novel medicines.  

However, the number of patients enrolled in the clinical trials is very limited. Simple descriptive calculation of the potential cost savings from clinical trials is not meaningful at all to address the financial burden of medication with expensive novel medicines of cancer patients.

Most importantly, the authors did not capture the characteristics of the enrolled and benefited patients in terms of their socio-economoc, demographic status, which might help to enable more indepth analyses of the meaningfulness of the potential cost savings from enrolling into the clinical trials to access to free medications. 

Reviewer 2 Report

Comments and Suggestions for Authors

I would like to commend the authors for their thorough research in the underexplored field of pharmaco-economics. The article evaluates the characteristics of clinical trials on cancer, specifically solid tumors, and the potential for cost savings in drug expenses, thanks to private sponsorship, in a population of adult patients over a 10-year period (2010-2019).

The methodology is sound and well-documented, with data that aligns with the findings. It appears evident that providing medications for free can result in savings for the hospital entity, a conclusion well demonstrated in this study despite the heterogeneity of the research sources.

Perhaps an update on the prices by deflating them, using relative prices, or any other economic indicator to allow for equal expense comparisons over the time series could have been considered. Yet, these are considerations the Authors might address in future research, including fixed costs for personnel, furniture, etc., to render the results more realistic, possibly leveraging analytical accounting, for example.

Finally, they demonstrate the importance of such collaborations for early access to innovation and contribute knowledge that benefits clinical care practices beyond alleviating the economic burden on the healthcare system through the impact of free drug provision in research and the potential pharmaceutical expenditure avoided from the health system's reimbursement perspective.

Reviewer 3 Report

Comments and Suggestions for Authors

Congratulations to the authors for this very interesting work on the drug cost avoided by the performing of clinical trials. This is of high relevance in the actual contest of cost-saving in healthcare policies. The study was well performed. written and presented. I have only one curiosity that I would implement in the method to help readers at the international level to understand how is organize your healthcare system. Please, in the method section can you add a brief description of your healthcare system, and explain in which category your clinic has belonged (public, private, or mixed...) How is made up the cost of drugs? Who are the payers? There are any managed entry agreements?

Round 2

Reviewer 1 Report

Comments and Suggestions for Authors

As stated in the last round of comments, I do not see the added value of this analysis.  
